# Rubber Genotypes with Contrasting Drought Factor Index Revealed Different Mechanisms for Drought Resistance in *Hevea brasiliensis*

**DOI:** 10.3390/plants11243563

**Published:** 2022-12-16

**Authors:** Andi Nur Cahyo, Rudi Hari Murti, Eka Tarwaca Susila Putra, Fetrina Oktavia, Sigit Ismawanto, Pascal Montoro

**Affiliations:** 1Indonesian Rubber Research Institute, Sembawa, Banyuasin 30953, Indonesia; 2Department of Agronomy, Faculty of Agriculture, Universitas Gadjah Mada, Yogyakarta 55281, Indonesia; 3CIRAD, UMR AGAP Institut, F-34398 Montpellier, France; 4CIRAD, INRAE, UMR AGAP Institut, Institut Agro, University Montpellier, F-34398 Montpellier, France

**Keywords:** drought factor index, drought resistance, *Hevea brasiliensis*, transpiration, leaf gas exchange

## Abstract

It is predicted that drought will be more frequent and sustained in the future, which may affect the decline of rubber tree production. Therefore, it is critical to research some of the variables related to the drought-resistance mechanism of the rubber tree. As a result, it can be used to guide the selection of new rubber drought-resistance clones. The goal of this study was to identify drought-resistance mechanisms in rubber clones from the high drought factor index (DFI) group using ecophysiological and biochemical variables. The treatments consist of two factors, namely water deficit and contrasting clones based on the DFI variable. The first factor consisted of three levels, namely normal (fraction of transpirable soil water (FTSW) > 0.75), severe water deficit (0.1 < FTSW < 0.20), and recovery condition (FTSW > 0.75 after rewatering). The second factor consisted of seven clones, namely clones G239, GT1 (low DFI), G127, SP 217, PB 260 (moderate DFI), as well as G206 and RRIM 600 (high DFI). RRIM 600 had the highest DFI among the other clones as a drought-tolerance mechanism characteristic. Furthermore, clones RRIM 600, GT1, and G127 had lower stomatal conductance and transpiration rate than drought-sensitive clone PB 260. As a result, as drought avoidance mechanisms, clones RRIM 600, GT1, and G127 consume less water than clone PB 260. These findings indicated that clone RRIM 600 was a drought-resistant clone with drought tolerance and avoidance mechanisms.

## 1. Introduction

Climate change phenomenon is predicted to increase the frequency and duration of dry periods [1]. This condition is called drought when it results in plant injury [2]. In addition to influencing the growth and yield of plants, ecophysiological variables are affected by drought, for example, osmotic and turgor pressure, conductance of stomata, photosynthesis, transpiration, respiration, and antioxidant activity [3,4,5,6,7]. Among the effects of drought on plant ecophysiological processes, changes in stomatal conductance are the most prominent [8]. The decrease in stomatal conductance resulted in protective effects such as a decrease in transpiration rate, allowing plants to save water and improve water use efficiency [7,9]. On the contrary, the closure of stomata as a result of drought resulted in a decrease in CO_2_ assimilation rate, resulting in excessive light exposure, over-reduction of the PSII (photosystem II) photosynthesis reaction center, and the production of ROS (Reactive Oxygen Species). The accumulation of ROS can damage leaf cell tissue and leads to oxidative stress [10]. As a result of the drought, the plant growth rate and yield decreased.

Plants can adapt to drought conditions through various mechanisms such as drought escape, avoidance, tolerance, and recovery [11,12]. Plant drought-resistance mechanisms include, for example, reducing stomatal conductance to achieve a low transpiration rate (avoidance mechanism). As a result, water is available in plant cells for a longer period, and plant dehydration can be delayed. Furthermore, the mechanism of plant adaptation under drought conditions can be achieved by increasing the concentration of ROS-neutralizing substances such as enzymatic and non-enzymatic antioxidants (tolerance mechanism). Catalase (CAT), superoxide dismutase (SOD), and peroxidase (POD) are ROS-scavenging enzymes [7], while ascorbic acid, glutathione and NADPH are antioxidant metabolites [13]. SOD converts O_2_^−^ to H_2_O_2_. CAT participates in the conversion of H_2_O_2_ into H_2_O and O_2_ [14]. In addition, the ascorbate peroxidase (APX) converts H_2_O_2_ into H_2_O and MDHA (monodehydroascorbate) [15]. Furthermore, ascorbic acid is an important substrate for ROS detoxification. It is the electron source required by APX to convert H_2_O_2_ to H_2_O and O_2_ [13,16].

ROS can damage leaf cell tissue, including PSII, if these ROS-neutralizing substances fail to suppress ROS formation. PSII activity can be measured using chlorophyll-*a* fluorescence (CF). As a result, CF can be used to assess the overall physiological status of plants as a result of drought. Performance Index (PI) is a CF variable that reflects plant vitality [17]. Furthermore, by monitoring this integrative variable under normal, moderate, and severe drought conditions, we can calculate the drought factor index (DFI) to screen plants for drought stress tolerance [3,18]. The higher the DFI value, the more tolerant plants are to drought stress [19].

Because it reflects the change in plant vitality (PI) under normal, moderate, and severe water deficit conditions, DFI is associated with the tolerance mechanism. DFI represents the relative reduction in PI caused by drought during a drought stress period [17]. Drought-tolerant genotypes should be able to tolerate drought stress for longer periods of time than drought-sensitive genotypes, according to the DFI principle [20]. A low DFI indicates a greater decrease in PSII activity due to drought. A genotype’s drought tolerance is also indicated by a higher DFI [19]. DFI has been used successfully to sort the drought tolerance level of some genotypes of some species, including *Sesamum indicum* [21], *Cicer arientinum* L. [22], *Hordeum vulgare* L. [20], and *Elaeis guineensis* [23].

The DFI of perennial tree species, including *Hevea brasiliensis*, is poorly understood. *H. brasiliensis* is a natural rubber-producing tree species classified as a C3 plant, making it more susceptible to drought than a C4 plant [24]. According to some studies, drought reduced the growth and yield of the rubber tree [3,25,26]. Only one drought-tolerant rubber clone, RRIM 600, has been identified and is being used in dry areas of Thailand and India [27,28,29,30].

DFI analysis on a biparental population of PB 260 × SP 217 and several control clones (RRIM 600, PB 260, SP 217, and GT1) revealed three distinct groups of clones in drought response, namely a high DFI group (clones RRIM 600 and G206), a moderate DFI group (clones PB 260, SP 217, and G127), and a low DFI group (clones GT1 and G239) [31]. We characterized these selected contrasting rubber clones for DFI at the ecophysiological and biochemical levels. This study led to identifying drought-resistance mechanisms for each studied genotype and revealed that the well-known drought-resistant clone RRIM 600 exhibited two distinct drought adaptation mechanisms. DFI was, therefore, a robust integrative parameter useful for screening genetic populations for drought tolerance.

## 2. Results

### 2.1. Drought Factor Index 

DFI is an integrative index that measures a plant’s ability to withstand a water deficit. DFI was calculated based on the change in PI in three water deficit levels, namely normal, moderate, and severe. As a result, the DFI value was associated with the integrative index value of the drought-tolerance mechanisms. Figure 1A depicts the change in PI values of the contrasting clones based on the DFI value.

Figure 1A shows that the drop in PI during the transition from normal to severe water deficit was smaller for rubber clones with higher DFI than for clones with lower DFI. The more stable PI values of rubber clones included in the high DFI group (determined in the previous study) during normal, moderate, and severe water deficit conditions resulted in a high DFI value in this study, as shown in Figure 1B. As a result, the findings of this study confirm the findings of a previous study conducted by Cahyo et al. [31].

### 2.2. Analysis of Variance and Principal Component Analysis of Some Ecophysiological and Biochemical Variables Related to Water Deficit

Water deficit and clone factors had a significant effect on all variables observed in this research. Whereas the interactions between water deficits and the clones were significant for variables *gs* (stomatal conductance), *E* (transpiration rate), VPD (vapor pressure deficit), and POD (peroxidase) enzyme activity (Table 1). Furthermore, principal component analysis (PCA) was used to identify important factors influencing the observed variables. PCA was carried out for normal, severe water stress and recovery conditions. It can generate a biplot graph of the observed variables and the treatment factor (rubber clones) that shows the relationship between them and the rubber clones. The two main axes F1 and F2, show 75.99, 76.02, and 70.33% for the respective conditions (Figure 2A–C).

Correlation coefficients greater than 0.8 were used to explain the graph. On the axis F1, VPD, *gs* and *E* were the most significant. High DFI genotypes (RRIM 600, 206) remain stable on the different graphs. By contrast, low and moderate DFI genotypes are not associated with the same factors for normal and other conditions (severe water deficit and recovery). Interestingly, two famous rubber clones are opposite, especially under stress conditions: clone RRIM 600 was associated with VPD whatever the conditions, while clone PB 260 was associated with high *gs* and *E*.

Furthermore, in severe water deficit conditions, the opposition of clones G206 and GT1 on the F2 axis could be explained by the variable POD (0.909). The low POD was linked to clone RRIM 600. The high POD, on the other hand, was associated with clone G206.

### 2.3. Effect of Interaction between Water Deficit and Clone on Several Ecophysiological and Biochemical Variables

The effect of interaction between water deficit and clone on *gs, E,* VPD, and POD is presented in Figure 3A–D, respectively.

Figure 3A,B shows that *gs* and *E* decreased during water deficit for all genotypes and then increased at a similar level with prior stress after rewatering. The decrease in *gs* (0.01–0.02 mol H_2_O m^−2^ s^−1^) and *E* were comparable (1–2 mmol H_2_O m^−2^ s^−1^). Furthermore, the high level of *gs* and *E* can be observed for any DFI level. It demonstrated that the levels of *gs* and *E* are unrelated to the DFI. Furthermore, Figure 3C shows that all the clones had the same VPD in normal conditions. A similar situation occurred when there was a water deficit. Except for clones PB 260 and GT1, which had lower VPD than clone G239, there was no significant difference in VPD. At recovery, the VPD of clones PB 260, G239, and G206 were significantly lower than those of clones SP 217, GT1, RRIM 600, and G127. In addition, Figure 3D shows that a lack of water caused a significant increase in POD in clones G206 and SP 217. In the recovery condition, POD increased significantly for clone RRIM 600. The low DFI group had the lowest POD enzyme activity when water was scarce.

## 3. Discussion

Rubber plants are stressed during the dry season due to water shortages. To cope with water shortage, several strategies are used by rubber plants, namely tolerance and avoidance mechanisms. Among several variables observed in this research, DFI is a variable associated with tolerance mechanisms because it reflects the ability of the plant to maintain the functionality of PS2 under drought conditions, whereas the avoidance mechanism is linked to the plant’s ability to maintain plant water status via stomatal conductance and transpiration rate adjustment. As a result, this section discusses some tolerance mechanism variables, such as DFI in relation to PI and POD enzyme activity, as well as avoidance mechanism variables, such as stomatal conductance, transpiration rate, and VPD, in detail. The expected mechanism of each clone is also discussed in this section.

This research used the PCA biplot analysis to determine the correlation between variables and their association with rubber clones. The PCA was used because it can determine the important components affecting the observed variables. Furthermore, it can generate a biplot graph of observed variables and the treatment factor (rubber clones); hence the association between observed variables and rubber clones can be determined. The PCA revealed that RRIM 600 as a control for drought-resistant clones and PB 260 as a control for the drought-sensitive clone were located in opposite directions, especially in severe water deficit and recovery conditions. The PCA also revealed that under severe water deficit and recovery conditions, clone PB 260 was characterized by high *gs* and *E*. On the contrary, under the same water deficit level, clone RRIM 600 was characterized by low *gs* and *E*. The low *gs* and *E* under water deficit and recovery conditions indicated that clone RRIM 600 could withstand drought conditions through avoidance mechanisms, besides other mechanisms that are discussed in the discussion section.

### 3.1. Tolerance Mechanism 

The highest DFI variable was found in clones RRIM 600 (control for drought-resistant clones) and G206, while the lowest DFI variable was found in clones GT1 and G239. PB 260, a widely known drought-sensitive clone [8,32,33], was classified as having moderate DFI based on its eco-physiological, biochemical, growth, and yield characteristics (Figure 1B). This result was consistent with previous research [31]. The high DFI of clones RRIM 600 and G206 suggested that they could keep PI stable in normal, moderate, and severe water deficit conditions. The PI in severe water deficit conditions for clones GT1 and G239, on the other hand, was reduced to 75% of normal condition PI values (Figure 1A). 

The ability of high DFI rubber clones to maintain high PI in the presence of severe water deficit was linked to ROS scavenging enzymes such as POD. During water deficiency, the higher DFI group generated more POD enzyme activity. On the contrary, the lowest POD enzyme activity was observed in the group with the lowest DFI (Figure 3D). POD enzyme is required for the conversion of H_2_O_2_ to water [14,34,35]. As a result, the clone with the lowest POD enzyme activity of the lowest DFI group significantly affected the reduction of PI in the water deficit condition (Figure 1A). The high POD enzyme activity was required to compensate for the high H_2_O_2_ content caused by the water deficiency condition. If not converted to water, accumulation of H_2_O_2_ during water deficit conditions influenced the increase of EL, indicating damage to the plant cell membrane. Hence the clones with low DFI were more prone to drought conditions compared to the clones with high DFI. 

Due to the impact of high POD enzyme activity, high DFI rubber clones could maintain the PI at a high level under severe water deficit conditions. Under water-stress conditions, the POD enzyme is responsible for scavenging H_2_O_2_, allowing cell membrane damage to be avoided [35]. At water-deficit conditions, the high POD enzyme activity resulted in a high PI. As a result, the biochemical variables and DFI were associated with the tolerance mechanism because it reflects plant vitality under water-stress conditions. Plant metabolism is enabled by the tolerance mechanism at low water potential [36].

Overall, the tolerance mechanism of rubber plants to withstand water deficit conditions is presented in Figure 4.

### 3.2. Avoidance Mechanism 

Plant adaptation to water deficit conditions can be both a tolerance and an avoidance mechanism. The avoidance mechanism is plant water-status homeostasis, which includes stomatal control to manage the transpiration rate [36]. In this study, when clones RRIM 600, SP 217, GT1, and G127 were compared to PB 260 as the control for drought-sensitive clones, low *gs* (Figure 3A) and *E* (Figure 3B) as the avoidance mechanism to preserve water inside plant tissues occurred, especially in recovery conditions. The recovery condition represents the real-world situation in the field when rainfall is not evenly distributed over a month, and the rubber tree is subjected to a brief period of drought stress. The lower *E* of clones RRIM 600, SP 217, GT1, and G127 compared to clone PB 260 indicated that clones RRIM 600, SP 217, GT1, and G127 could keep the planting medium moist for a longer period of time than clone PB 260. In water-limited conditions, clones RRIM 600, GT1, and G127 survive for a longer period of time than PB 260. This also suggested that, in addition to the tolerance mechanism, clone RRIM 600 used an avoidance mechanism to survive in drought conditions.

Furthermore, Figure 2 shows that clone RRIM 600 had the opposite characteristics as clone PB 260 in all ecophysiological variables, particularly in severe water deficit (Figure 2B) and recovery (Figure 2C) conditions. It was also clearly demonstrated that PB 260 was associated with high *gs, E,* and low VPD in severe water deficit and recovery conditions. RRIM 600, on the other hand, was associated with low *gs, E,* and VPD. Lowering *gs* resulted in lower *E* [9,37,38]. When plants detect a water shortage at the root zone, they produce the ABA hormone. The ABA hormone was then translocated to the leaf as an indicator of a water deficit [39]. Because this hormone caused stomatal closure, *E* was reduced [40]. As a result, increasing *gs* resulted in increasing *E* and vice versa. 

The plant materials used in this research were produced by a budding technique using GT1 seedlings as the rootstocks, which might be had a high variability due to the high heterozygosity of rubber plants. However, because we used a fraction of transpirable soil water (FTSW) to determine the water deficit level, the effect of the variability of root properties on the inaccuracy of plant water deficit level determination for each treatment could be suppressed.

Furthermore, Figure 3A shows that in the absence of water, the *gs* of all clones were significantly lower than in the presence of water. For most of the clones, the *gs* in water deficit conditions were reduced by more than 50%. This condition increased the production of reactive oxygen species (ROS) above normal levels. Excess ROS in the leaves resulted in oxidative stress and leaf cell damage [10]. Furthermore, when water was added to the plant to achieve recovery conditions, the *gs* of all clones recovered to the same level as the *gs* in normal conditions, except for clone PB 260. When PB 260 was in recovery condition, its *gs* doubled when compared to normal condition. This condition could also occur due to a decrease in cuticle permeability caused by a water deficit, resulting in a drop in *E* as seen in the tea plant [41]. While rewatering treatment resulted in a decrease in cuticle barrier (intercellular wax coverage) compared to water deficit and normal conditions, *E* increased [42]. The increased transpiration following rewatering made clone PB 260 more susceptible to water stress. This trait may make clone PB 260 more vulnerable during the transition from dry to wet periods. However, the high *E* after rewatering indicated that clone PB 260 recovered the fastest from a water deficit condition. It implied that when water availability is sufficient to meet clone PB 260’s water requirements, clone PB 260’s productivity would recover faster than clone with low *E* after rewatering, such as GT1 [33].

### 3.3. Expected Mechanisms of some Rubber Clones to Withstand Water Deficit Conditions 

The drought-resistance expected mechanisms of rubber clones with contrasting DFI can be classified into three categories based on DFI and *E* variables, namely drought-sensitive, drought avoidance, and drought-tolerance mechanisms. Table 2 shows the expected resistance mechanism of the seven rubber clones with contrasting DFI. 

Table 2 shows that clones G239 and PB 260 were expected to be drought sensitive due to their low and moderate DFI levels, respectively, as well as their high E. Furthermore, due to the low *E*, clones GT1, G127, SP 217, and RRIM 600 were expected to have an avoidance mechanism. Furthermore, due to the high DFI, clones G206 and RRIM were expected to have drought-tolerance mechanisms. These findings revealed that, of all the rubber clones tested in this study, only RRIM 600, as a drought-resistant clone possessed both drought tolerance and drought avoidance mechanisms. 

Clone RRIM 600 reduced *gs* under drought conditions, resulting in an *E* reduction. As a result, water consumption could be reduced, and water availability could be preserved for a longer period (avoidance mechanism). Clone RRIM 600, on the other hand, could maintain PI as high as normal conditions under water deficiency conditions. Furthermore, because PI reflects plant vitality, the low leaf cell damage was also indicated by high PI. The high PI in a water-stressed environment resulted in a high DFI. As a result, DFI could be associated with a tolerance mechanism.

### 3.4. Summary of the Findings

DFI is a robust parameter to select a drought-tolerant plant, including rubber. The higher DFI indicated the higher tolerance of rubber clones to withstand water deficit conditions. The highest DFI was found for the well-known drought-resistant clone RRIM 600 that could maintain stable PI as well as stable PS2 functionality during normal and water deficit conditions as an expression of tolerance mechanism.

Furthermore, when clones RRIM 600, SP 217, GT1, and G127 were compared to PB 260 as the control for drought-sensitive clones, low *gs* and *E* were found. Clones RRIM 600, SP 217, GT1, and G127 consume less water than clone PB 260 due to their low *gs* and *E*. This fact indicated that these three clones were more efficient in using water than clone PB 260 and thus could withstand water deficit conditions for a longer period of time than clone PB 260. As a result, clones RRIM 600, SP 217, GT1 and G127 were expected to use an avoidance mechanism to cope with a water deficit.

Interestingly, this study revealed that the drought-resistant clone RRIM 600 exhibits both drought tolerance and adaptative avoidance mechanisms. These findings will be confirmed by ecophysiological studies of mature trees grown in small-scale clone trials in order to validate the use of DFI on juvenile plant material as a robust integrative parameter for screening genetic populations for drought tolerance.

## 4. Materials and Methods

### 4.1. Experimental Site and Plant Materials

This study was conducted from December 2021 to February 2022 in the greenhouse of the Faculty of Agriculture, Universitas Gadjah Mada, Yogyakarta, Indonesia, located at 7°46′5.3256″ S and 110°22′55.236″ E with an altitude of 144 m above sea level.

The plant materials consisted of three groups of clones with contrasting DFI [31]. The three contrasting groups of clones were clone numbers G239 and GT1 (low DFI), G127, SP 217, and PB 260 (moderate DFI), as well as G206 and RRIM 600 (high DFI). Clone numbers G239, G127, and G206 were derived from crossing between clones PB 260 and SP 217. In contrast, clone GT1 was a primary clone originating from Indonesia and used as control. While PB 260 and RRIM 600 were controls for drought-sensitive and -resistant clones, respectively. The scions of these clones were budded onto GT1 seedlings as the rootstocks. The budding activity was conducted in August 2021 when the rootstock at the ground nursery was six-month-old. The budded plant materials were then planted in polybags (size 12 × 25 cm) in September 2021. The water deficit treatment was started on December 2021 when the leaves reached a mature age, indicated by the dark green color of the leaves with a chlorophyll index was more than 50 index units.

### 4.2. Experimental Design and the Treatments

This research was conducted using a Completely Randomized Design (CRD). The treatments consist of two factors, namely water deficit level and rubber genotypes/clones. The water deficit level was determined based on the fraction of transpirable soil water (FTSW). The first factor (water deficit level) consisted of three levels, namely normal condition (FTSW > 0.75), severe water deficit (0.1 < FTSW < 0.20), and recovery condition (FTSW > 0.75 after rewatering). The second factor (rubber clones) consisted of three levels, namely low DFI (G239 and GT1), moderate DFI (G127, SP 217, and PB 260), and high DFI (G206 and RRIM 600). These treatments were replicated five times, and each replication consisted of one unit of the plant. 

To start the research, water deficit levels treatment was given once the leaves reached the maximum size with dark green color (two months old leaves). The FTSW for water deficit levels determination was calculated by plotting the NTR (normalized transpiration rate) value in the NTR and FTSW relation curve with the formula [43]:NTR = 2/[1 + exp(−14 * FTSW)] − 1(1)

The NTR was the normalized ratio of transpiration rate (TR). The TR was determined by dividing the daily transpiration of each individual rubber plant material of a given genotype which was treated by water deficit, by the average daily transpiration of the control treatment of that genotype. Then, the NTR was determined by dividing the TR value over time by the average of the first three-day TR values when the availability of water was not limited under the water deficit treatment [44]. Daily transpiration was determined by calculating the difference between the weight of the covered polybag of the individual plant on two consecutive days. This activity was conducted from the beginning of the water deficit treatment until the plant reached severe water deficit conditions.

### 4.3. Measurement of CF and Calculation of DFI

The CF variable (PI) was measured using a Pocket PEA [45]. This device uses 3500 µmol m^−2^ s^−1^ of saturating light pulses for 1 s [45]. Based on the previous research by Cahyo et al. [46], a dark adaptation time of one hour was used prior to the CF measurement to get the correct F_0_ (minimum fluorescence) level. This measurement was conducted at 07.30 a.m. at the same point of the leaf when the water deficit level of water deficit treated plants reached normal, moderate, and severe water deficit conditions as calculated using the NTR and FTSW methods. 

Furthermore, DFI is calculated as follows [20]: DFI = logA + 2 logB(2)
Note:
A = PI relative = PI at moderate water-deficit condition/PI in normal conditionsB = PI relative = PI at severe water-deficit condition/PI in normal conditions

The measurement of PI was conducted at a certain range of FTSW based on the research reported by Lestari et al. [47] and Sanier et al. [8], namely: normal condition, (FTSW > 0.75), moderate water-deficit condition (FTSW = 0.35–0.45), and severe water-deficit condition (FTSW = 0.1–0.2).

### 4.4. Measurement of Ecophysiological and Biochemical Variable

#### 4.4.1. Measurement of Ecophysiological Variables

The ecophysiological variables observed in this study, namely *gs* (mol H_2_O m^−2^ s^−1^), *E* (µmol H_2_O m^−2^ s^−1^), and VPD (kPa). These variables were measured using Portable Photosynthesis System Licor Li-6400 XT [48] at the end of the experiment when the plants reached normal, severe water deficit, and recovery condition. This measurement was conducted from 11.00 a.m.–02.30 p.m. when the sunlight was optimum for the photosynthesis process with a quantum flux of 7955 µmol m^−2^ s^−1^. Furthermore, the concentration of CO_2_ applied in the leaf chamber was 415 µmol/mol.

#### 4.4.2. Measurement of Biochemical Variables

The biochemical variables observed in this study, namely EL (%), H_2_O_2_ content (ppm), SOD, and POD enzyme activity (unit mg^−1^).

EL

EL was determined by cutting 5 mm length of 100 mg fresh leaf samples and placing it in the tubes containing 10 mL distilled deionized water. The tubes are covered by plastic caps and placed in a water bath. These tubes are maintained at a temperature of 32 °C. Initial medium electrical conductivity (EC1) is measured after 2 h. After that, the samples were autoclaved at 121 °C for 20 min to completely kill the tissues and release all electrolytes. After that, samples are cooled to 25 °C, and the final electrical conductivity (EC2) is measured. The electrolyte leakage EL was calculated using the formula EL = EC1 : EC2 × 100 [49]. 

H_2_O_2_

H_2_O_2_ was measured spectrophotometrically after a reaction with Potassium Iodate (KI). The reaction mixture consisted of 0.5 mL 0.1% TCA (trichloroacetic acid) leaf extract supernatant, 0.5 mL of 100 mM K-phosphate buffer and 2 mL reagent (1 M KI *w*/*v* in fresh double-distilled water). The blank probe consisted of 0.1% TCA without leaf extract. The reaction is developed for 1 h in the dark condition, and the absorbance is measured at 390 nm. Hydrogen peroxide content is calculated using a standard curve prepared with determined concentrations of H_2_O_2_ [50].

SOD and POD enzyme activity

The SOD (EC1.15.1.1) enzyme activity was determined using the method reported by Marklund and Marklund (1974) [51]. To determine the activity of SOD, a fresh matter of rubber leaves (1 g) was grounded in liquid nitrogen and centrifuged at 4 °C for 30 min at 45,000 rpm. 200 µL of phosphate buffer (pH 6.5), 1000 µL of Tris-EDTA (pH8.2) and 10 µL of pyrogallol (0.2 mM) then was added into 100 µL of the supernatant. The reaction was monitored at 420 nm to measure the decrease of the absorbance per 30 s. POD (EC 1.11.1.7) enzyme activity was determined by using the same supernatant prepared for SOD activity calculation. The supernatant (100 µL) was mixed with distilled water (1000 µL), potassium phosphate buffer pH 6 (160 µL), hydrogen peroxide (80 µL), and pyrogallol (180 µL). The mixture was measured at 420 nm for 90 s with an interval of 30 s.

### 4.5. Statistical Analysis

A completely randomized design analysis of variance with a 95% confidence interval was used to analyze the effect of the treatments on the observed variables. If there was a significant difference between the mean, a further test was conducted by using Least Significant Difference (LSD) test. This statistical analysis was conducted using the R software. Furthermore, PCA was employed to determine correlations between observed variables and their association with the clones. This analysis was conducted using XLSTAT 2022 [52].

## 5. Conclusions

Clone RRIM 600, as the control for the drought-resistant clone, had the highest DFI among other clones. Furthermore, clones RRIM 600, SP 217, GT1, and G127 had lower stomatal conductance (*gs*) and transpiration rate (*E*) compared to clone PB 260 as the control for the drought-sensitive clone. Hence, clones RRIM 600, SP 217, GT1, and G127 consume less water than clone PB 260. These results indicated that clone RRIM 600 was a drought-resistant clone based on drought tolerance and its avoidance mechanisms. Therefore, the selection of drought-resistant plants in the future should consider drought tolerance and avoidance mechanism by using DFI and *E,* for example, as the observation variables.

## Figures and Tables

**Figure 1 plants-11-03563-f001:**
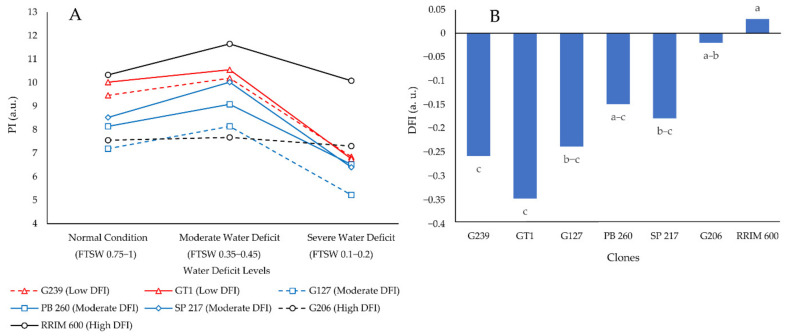
Fluctuations in PI (**A**) and DFI (**B**) of seven different rubber clones under normal, moderate, and severe water deficit conditions. a.u. stands for arbitrary unit. The different letters represent significantly different means as determined by the Least Significant Difference test at 95% confidence.

**Figure 2 plants-11-03563-f002:**
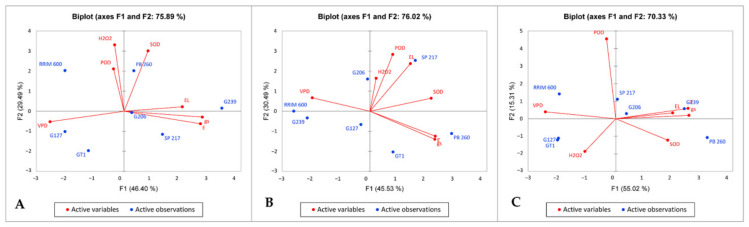
The results of the biplot analysis of all observed variables and rubber clones under three different water deficit conditions, namely: normal condition (**A**), severe water deficit condition (**B**), and recovery condition (**C**). Note: *gs*—stomatal conductance (mol H_2_O m^−2^ s^−1^), *E*—transpiration rate (mmol H_2_O m^−2^ s^−1^), VPD—vapor pressure deficit (kPa), EL—electrolyte leakage (%), H_2_O_2—_hydrogen peroxide content (ppm), SOD—superoxide dismutase enzyme activity (unit mg^−1^), POD—peroxidase enzyme activity (unit mg^−1^).

**Figure 3 plants-11-03563-f003:**
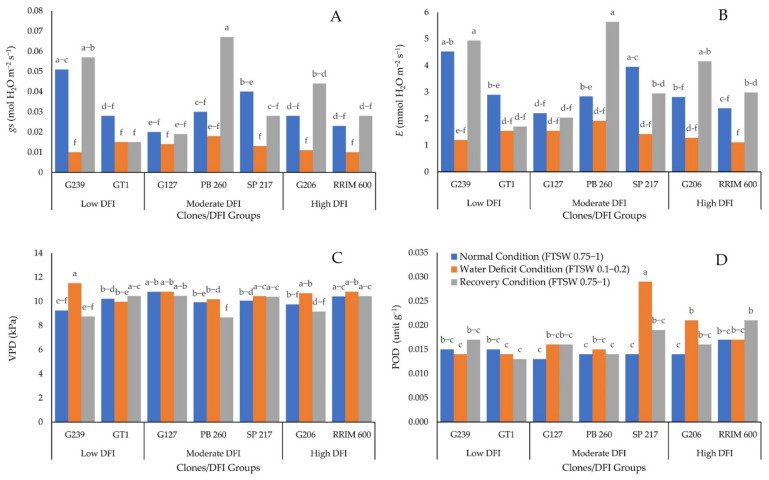
Effect of combination between clone and water deficit level on: (**A**) *gs*, (**B**) *E*, (**C**) VPD, and (**D**) POD enzyme activity. The different letters indicate significantly different means determined by the Least Significant Difference test at a 95% confidence interval.

**Figure 4 plants-11-03563-f004:**
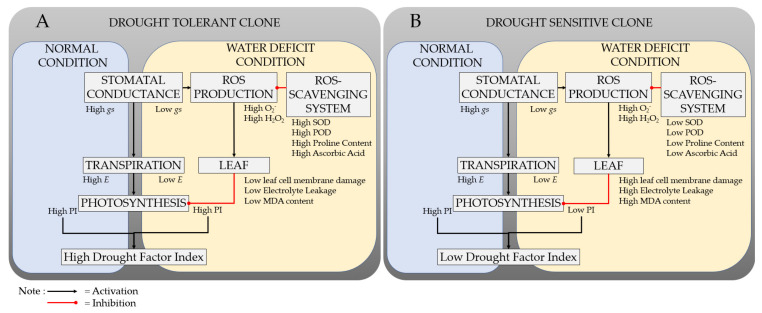
Response mechanisms of drought-tolerant (**A**) and drought-sensitive clone (**B**) to normal and water deficit conditions.

**Table 1 plants-11-03563-t001:** The results of the analysis of variance of several ecophysiological and biochemical variables as the effect of water deficit and clone treatment factors. Note: F values followed by ***, **, and * are significant at α = 0.001, 0.01, and 0.05, respectively. DF—degree of freedom, *gs*—stomatal conductance (mol H_2_O m^−2^ s^−1^), *E*—transpiration rate (mmol H_2_O m^−2^ s^−1^), VPD—vapor pressure deficit (kPa), EL—electrolyte leakage (%), H_2_O_2—_hydrogen peroxide content (ppm), SOD—superoxide dismutase enzyme activity (unit mg^−1^), POD—peroxidase enzyme activity (unit mg^−1^).

Source of Variation	DF	F Value													
		*gs*		*E*		VPD		EL		H_2_O_2_		SOD		POD	
Water Deficit	2	16.006	***	21.986	***	8.171	***	5.136	**	8.577	***	30.539	***	3.804	*
Clone	6	3.388	**	3.385	**	2.811	*	4.868	***	3.37	**	2.836	*	2.347	*
Water Deficit × Clone	12	1.928	*	2.098	*	2.069	*	0.526		1.06		1.287		1.978	*
Residuals	81														

**Table 2 plants-11-03563-t002:** Drought-resistance expected mechanism of seven rubber clones with contrasting DFI.

Rubber Clone	Drought Factor Index	Transpiration Rate	Drought-Resistance Expected Mechanism
G239	Low	High	Drought Sensitive
GT1	Low	Low	Drought Avoidance
G127	Moderate	Low	Drought Avoidance
PB 260	Moderate	High	Drought Sensitive
SP 217	Moderate	Low	Drought Avoidance
G206	High	Moderate	Drought Tolerance
RRIM 600	High	Low	Drought Tolerance and Avoidance

## Data Availability

The data presented in this study are available within the article.

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
