# Peer review of "Rubber Genotypes with Contrasting Drought Factor Index Revealed Different Mechanisms for Drought Resistance in *Hevea brasiliensis"

_plants, 2022, doi:10.3390/plants11243563_

Round 1

Reviewer 1 Report

Comments are in the attached file.

Author Response

Comments/ Suggestions

Original version

Revised version

Line number in the revised version

Lines 14-15, Your ideas here can be better put into the following sentence “It is predicted that drought will be more frequent and sustained in the future, which may affect the decline of rubber tree production.”, as the ‘occurrence’ of something cannot be ‘longer’.

Drought occurrence is predicted to be more frequent and longer in the future that can have effect on rubber tree production decline.

It is predicted that drought will be more frequent and sustained in the future, which may affect the decline of rubber tree production.

Lines 14-15

Line 18, define the initial “DFI” here instead of in line 20.

… high DFI group …

… high drought factor index group (DFI)…

Line 18

Line 18, define the initial “DFI” here instead of in line 20.

… The treatments consist of two factors, namely water deficit and contrasted clones based on drought factor index (DFI) variable …

… The treatments consist of two factors, namely water deficit and contrasting clones based on DFI variable …

Line 20-21

Line 21, define ‘FTSW’

… normal (FTSW >0.75), …

… normal (fraction of transpirable soil water (FTSW) >0.75), …

Lines 21-22

Line 34, phenomena → phenomenon

Climate change phenomena …

Climate change phenomenon …

Line 36

PSII

PSII (photosystem II)

Line 46

Line 45, leads to → and leads to

Accumulation of ROS can damage leaf cell tissue leads to oxidative stress [10].

Accumulation of ROS can damage leaf cell tissue and leads to oxidative stress [10].

Lines 46-47

Lines 54-61, there are some repeats of same thing, rewrite.

Catalase (CAT), superoxide dismutase (SOD), and peroxidase (POD) are enzymatic antioxidants [7]. In contrast, non-enzymatic antioxidants such as ascorbic acid (Vitamin C) [13]. SOD is a ROS-scavenging enzyme that converts O2- formed during drought to H2O2. CAT, on the other hand, is an enzyme that participates in the conversion of H2O2 to H2O and O2 [14]. In addition, the APX enzyme converts H2O2 to H2O and MDHA (monodehydroascorbate) [15]. Furthermore, ascorbic acid is an important substrate for ROS detoxification. It is the electron source required by APX to convert H2O2 to H2O and O2 [13,16].

Catalase (CAT), superoxide dismutase (SOD), and peroxidase (POD) are ROS-scavenging enzymes [7], while ascorbic acid,  glutathione and NADPH are antioxidant metabolites [13]. SOD converts O2- to H2O2. CAT participates in the conversion of H2O2 into H2O and O2 [14]. In addition, the ascorbate peroxidase (APX) converts H2O2 into H2O and MDHA (monodehydroascorbate) [15].

Lines 55-62

Lines 135-136, explain the result clearly.

… respectively. Correlation coefficients greater than 0.8 were used to explain the graph. On the axis F1, …

… respectively. On the axis F1, …

Lines 134-135

Lines 158-171, for explaining the result of Figure3, it is not necessary to separate the explanation into different paragraphs.

Lines 158-171 in the original version consist of three paragraphs.

Lines 154-166 in the revised version consist of one paragraph.

Lines 154-166

Line 173-182, rewrite the first paragraph. In discussion, you discuss your results and present what is your hypothesis, do not need to say ‘The mechanisms of adaptation of some rubber clones to water deficit conditions were hypothesized in this study’. The first sentence is not clear, perhaps you want to say ‘Rubber plants are under stress in dry season due to water shortage’

The drought season caused rubber plant stress due to a lack of water. The mechanisms of adaptation of some rubber clones to water deficit conditions were hypothesized in this study. Tolerance and avoidance mechanisms were among the mechanisms. As the integrative variable of drought tolerance parameters, the tolerance mechanism is associated with DFI. Whereas the avoidance mechanism is linked to the plant's ability to maintain plant water status via stomatal conductance and transpiration rate adjustment. As a result, this section discusses some tolerance mechanism variables, such as DFI in relation to PI and POD enzyme activity, as well as avoidance mechanism variables, such as stomatal conductance, transpiration rate, and VPD, in detail. The expected mechanism of each clone is also discussed at the end of the discussion section.

Rubber plants are stressed during the dry season due to water shortage.  To cope with water shortage, several strategies are used by rubber plant, namely tolerance and avoidance mechanisms. Among several variables observed in this research, DFI is a variable that associated with tolerance mechanism, because it reflects ability of plant to maintain the functionality of PS2 under drought condition. Whereas the avoidance mechanism is linked to the plant's ability to maintain plant water status via stomatal conductance and transpiration rate adjustment. As a result, this section discusses some tolerance mechanism variables, such as DFI in relation to PI and POD enzyme activity, as well as avoidance mechanism variables, such as stomatal conductance, transpiration rate, and VPD, in detail. The expected mechanism of each clone is also discussed in this section.

Lines 168-177

You hypothesized that ROS, SOD and POD are involved in rubber plant response to drought stress and two mechanisms, tolerance and avoidance, are hypothesized. Are they involved in both mechanisms? If yes, please explain how and the difference.

ROS, SOD and POD are involved in rubber plant response to drought stress. Furthermore, two mechanisms, tolerance and avoidance, are hypothesized.

ROS, SOD, and POD are just involved in tolerance mechanism. It is explained in the sentence below (Lines 217-218)

As a result, the biochemical variables and DFI were associated with the tolerance mechanism because it reflects plant vitality under water-stress conditions.

217-218

Line 242, FTSW needs to be defined when it appears the first time both in Abstract and the main body of the manuscript.

… we used FTSW to determine …

… we used fraction of transpirable soil water (FTSW) to determine …

Lines 250-251

Lines 311-312, you use seven genotypes or clones of rubber tree in your study, according to your description, these seven clones were assorted into three groups, tolerant, moderate and sensitive ones, therefore, the second factor may consist of three levels instead of seven.

The second factor (rubber genotypes and clones) consisted of seven levels, namely gen-otypes number G239, G127, G206, PB 260, SP 217, GT1, and RRIM 600.

The second factor (rubber clones) consisted of three levels, namely low DFI (G239 and GT1), moderate DFI (G127, SP 217, and PB 260), and high DFI (G206 and RRIM 600).

Lines 336-338

Lines 349-350 and other lines, VPD does not need to be defined all across the manuscript.

EL and other acronyms also need to define the first time they appear and do not need to define all across the manuscript.

… and vapor pressure deficit (VPD) (kPa).

EL

Electrolyte leakage

… and VPD (kPa).

Electrolyte Leakage (EL)

EL

Line 374

Line 124 (Table 1)

Lines 361, 362

Reviewer 2 Report

In the present manuscript "Rubber Genotypes with Contrasting Drought Factor Index Revealed Different Mechanisms for Drought Resistance in Hevea brasiliensis" authors have PCA and biplots to identify the variable and its significance in terms of drought stress.

Comments:

Please mention the importance of such studies and also justify the importance of such studies for the global audience in the Introduction section only.

Explain in details the use and benefits of performing PCA and biplots in identifying the variables for the drought study. And also discuss and how it’s better over any other such tests?

Use of abbreviation in the abstract itself may be avoided so that the message is clear from the abstract itself and it reach larger audiences.

 Discussion

A chapter has to be added in the discussion section which can summaries the overall findings based on the present study and its relevance to any other similar studies for future.

Page 2, line 47-49: Please rewrite to make it clear

Page 5, line 173: Please rewrite to make it clear

Page 8, line 302-303: Please rewrite to make it clear

Author Response

Comments/ Suggestions

Original version

Revised version

Line number in the revised version

Please mention the importance of such studies and also justify the importance of such studies for the global audience in the Introduction section only.

Drought-resistance mechanisms were identified for the clones, with clone RRIM 600 revealing two distinct drought adaptation mechanisms

This study led to identify drought resistance mechanisms for each studied genotype, and revealed that the well-known drought resistant clone RRIM 600 exhibited two distinct drought adaptation mechanisms. DFI was therefore a robust integrative parameter useful for screening genetic populations for drought tolerance

Lines 92-95

Explain in details the use and benefits of performing PCA and biplots in identifying the variables for the drought study. And also discuss and how it’s better over any other such tests?

Furthermore, PCA analysis was conducted for all variables to find out the influence of different variables (Figure 2). Figures 2A, 2B, and 2C show that the Biplot axes F1 and F2 could explain 75.89, 76.02, and 70.33% of the variables used in this study, respectively. With eigenvalues greater than one, the axes F1 and F2 could explain 46.40 and 29.49%; 45.53 and 30.49; and 55.02 and 15.31% for normal, severe water deficit, and recovery conditions, respectively. Correlation coefficients greater than 0.8 were used to explain the graph. On the axis F1, clone RRIM 600 was associated with low gs (0.959, 0.870, and 0.957 for normal, severe water deficit, and recovery condition, respectively) and E (0.937, 0.875, and 0.946 for normal, severe water deficit, and recovery condition, respectively). Clone PB 260, on the other hand, was associated with high gs and E in severe water deficit and recovery conditions. In terms of soil water efficiency, this result revealed an inverse relationship between RRIM 600 and PB 260.

Furthermore, principal component analysis (PCA) was used to identify important factors influencing the observed variables. PCA were carried out for normal, severe water stress and recovery conditions. It can generate a biplot graph of the observed variables and the treatment factor (rubber clones) that shows the relationship between the observed variables and the rubber clones. The two main axes F1 and F2 explain 75.99, 76.02, and 70.33% for the respective conditions (Figures 2A, 2B, and 2C). Correlation coefficients greater than 0.8 were used to explain the graph. On the axis F1, VPD, gs and E were the most significant. High DFI genotypes (RRIM 600, 206) remains stable on the different graphs. By contrast, genotypes with low and moderate DFI are not associated with the same factors for normal and other conditions (severe water deficit and recovery). Interestingly, two famous rubber clones are opposite especially under stress condition: clone RRIM 600 was associated with VPD whatever the conditions, while clone PB 260 was associated with high gs and E .

Lines 178-190

Use of abbreviation in the abstract itself may be avoided so that the message is clear from the abstract itself and it reach larger audiences

A lot of abbreviation in the abstract

Unnecessary abbreviations were discarded including E and gs

We also specified the name of FTWS

Lines 18-26